# Exploring the potential for introducing home monitoring of blood pressure during pregnancy into maternity care: current views and experiences of staff — a qualitative study

Lisa Hinton [iD],[1] James Hodgkinson [iD],[2] Katherine L Tucker [iD],[3]
Linda Rozmovits,[4] Lucy Chappell,[5] Sheila Greenfield,[2] Christine McCourt,[6]
Jane Sandall,[7] Richard J McManus [iD][3]

For numbered affiliations see end of article.

**Correspondence to**
Dr Lisa Hinton;
lisa.hinton@thisinstitute.cam.ac.uk

## ABSTRACT

**Objective** One in 20 women are affected by pre-eclampsia, a major cause of maternal and perinatal morbidity, death and premature birth worldwide. Diagnosis is made from monitoring blood pressure (BP) and urine and symptoms at antenatal visits after 20 weeks of pregnancy. There are no randomised data from contemporary trials to guide the efficacy of self-monitoring of BP (SMBP) in pregnancy. We explored the perspectives of maternity staff to understand the context and health system challenges to introducing and implementing SMBP in maternity care, ahead of undertaking a trial.

**Design** Exploratory study using a qualitative approach.

**Setting** Eight hospitals, English National Health Service.

**Participants** Obstetricians, community and hospital midwives, pharmacists, trainee doctors (n=147).

**Methods** Semi-structured interviews with site research team members and clinicians, interviews and focus group discussions. Rapid content and thematic analysis undertaken.

**Results** The main themes to emerge around SMBP include (1) different BP changes in pregnancy, (2) reliability and accuracy of BP monitoring, (3) anticipated impact of SMBP on women, (4) anticipated impact of SMBP on the antenatal care system, (5) caution, uncertainty and evidence, (6) concerns over action/inaction and patient safety.

**Conclusions** The potential impact of SMBP on maternity services is profound although nuanced. While introducing SMBP does not reduce the responsibility clinicians have for women's health, it may enhance the responsibilities and agency of pregnant women, and introduces a new set of relationships into maternity care. This is a new space for reconfiguration of roles, mutual expectations and the relationships between and responsibilities of healthcare providers and women.

**Trial registration number** NCT03334149.

## INTRODUCTION

Raised blood pressure (BP) in pregnancy affects around 1 in 10 women, almost half of whom develop pre-eclampsia, a major cause of maternal and perinatal morbidity, death and premature birth worldwide.[1 2] Diagnosis is made

### Strengths and limitations of this study

► Qualitative study to explore maternity staff perspectives on self-monitoring of blood pressure in pregnancy.
► Diverse sample includes voices from across multi-disciplinary maternity teams.
► Focus groups held across eight hospitals in the English National Health Services, including hospital and community midwives.

from monitoring BP and urine and symptoms at antenatal visits after 20 weeks of pregnancy.

Self-monitoring of BP (SMBP) in the general population has become more common and large trials have demonstrated its effectiveness in terms of BP control.[3] Since the 2011 UK national guidance for the general population with hypertension there has been a growing acceptance by clinicians of using SMBP values. Primary care physicians report an increasing use of SMBP to diagnose hypertension and for the ongoing management of hypertension to monitor BP control.[4–7]

But what are the implications of introducing SMBP for women during or after pregnancy, to improve the detection and management of gestational hypertension and pre-eclampsia? Pregnant women are different from the general population: BP can rise rapidly during pregnancy, and the problems associated with this (such as pre-eclampsia, eclampsia and stroke) are potentially serious. Pilot work in the UK suggests, with support from midwives and doctors, it is feasible and acceptable for women to monitor their BP and urine safely, potentially identifying hypertension earlier and controlling BP better.[8–10] SMBP during pregnancy is becoming common in some settings. A Canadian pilot study found

more than 60% of women diagnosed with non-proteinuric hypertension in pregnancy were already undertaking SMBP.[11] The Society of Obstetricians and Gynaecologists of Canada provides guidelines for SMBP in pregnancy and most Canadian obstetricians and primary care physicians used SMBP to assess for white coat hypertension (WCH),[12] but its place in other countries is not yet clear.[13]

While SMBP in pregnancy is already implemented in some UK settings,[14] there are no adequately powered randomised data from contemporary trials to guide the efficacy of such methods. Our early work exploring women's experiences of taking part in an SMBP pilot study[15] showed that most found their health professionals to be supportive, although in some cases self-monitoring results were ignored. An understanding of the attitudes of individuals and the organisation will be paramount in introducing new, and safe, models of care. Ahead of undertaking a large trial of BP monitoring in high-risk pregnancy to improve detection and monitoring of hypertension, we undertook a study to explore the perspectives of staff across the maternity care pathway to explore the context and health system challenges to introducing and implementing SMBP in maternity care.[10 16 17]

## METHODS

We carried out an intervention development phase in preparation for a large trial of SMBP in pregnancy, using qualitative methods, with the aim of understanding the views and prior experience of clinicians regarding such monitoring. This involved interviews and focus groups across eight participating teaching and urban general district hospitals that serve diverse populations (table 1). We conducted face-to-face interviews with a lead consultant obstetrician at each site (n=8) and a series of focus groups and interviews with a wide range of healthcare professionals.

### Sample/setting

We sought a broad pragmatic sample that included community and hospital-based midwives and obstetricians of different seniorities at each site (n=147). All focus groups and interviews took place at hospital or health centre sites, either as part of, or after, existing meetings or during shift breaks. It was usually a Research Midwife at each site who identified who might be available with some of the larger focus groups piggy-backing on existing meetings. Sampling was therefore convenience rather than purposive, though we ensured there was a representation of senior and junior obstetricians, and hospital and community midwives from each site. Most participants had not previously engaged in studies on SMBP. Focus group size ranged from 2 to 15 individuals (see table 1). Some focus groups had a mix of obstetricians and midwives participating, reflecting the multidisciplinary teams providing care to women during pregnancy. Although an obstetrican is the lead clinician for almost all women with additional risk factors in pregnancy, the midwife is the conduit of care throughout pregnancy, labour and the postnatal period.[18]

### Data collection

The interviews and focus groups were facilitated by two experienced, post-doctoral social science/nonclinical researchers (LH and JH) and audio-recorded for transcription and analysis. A topic guide was developed drawing on the literature, our prior work and in discussion with the multidisciplinary trial team (see Topic Guide, online supplemental file). Questions sought to explore issues around staff views on SMBP in pregnancy, the accuracy of home readings, how to manage care of women who present with home readings (particularly those discordant with clinic readings), the impact of home readings on clinical decision-making and workload, as well as practical issues around operationalising and integrating self-monitoring into current care pathways. Ethical approval and informed consent was obtained.

### Analysis

All interviews and discussions were transcribed for analysis, which was conducted in two phases. An initial rapid analysis was undertaken to guide trial development.[19] This used templates developed by JH and LH based on the topic guides and subsequently refined after a period of piloting. We conducted a rapid content analysis to identify evidence related to the a priori concepts and emergent themes that would then guide development of the study. This was discussed and refined with the wider research team. In a subsequent phase LH and JH, in discussion with the wider team, developed a coding frame and worked in parallel (with LR) to code and analyse the data.[20] Thematic analysis was undertaken to explore the anticipated and emergent themes and was then mapped back to the framework, supported by NVIVO.[21 22] LH and JH met frequently to discuss the results together and then with the wider team to confirm the credibility and dependability of the analysis. Data collection continued until data saturation was reached for the themes reported in this paper.[23] Interviews and focus groups were analysed as a single dataset, reflecting the multidisciplinary teams providing care to women with additional risk factors in pregnancy. Results are presented as staff views, with illustrative quotes presented in table 2.

### Patient and public involvement

This paper reports a substudy of a programme of intervention development work for the Blood pressure monitoring in high-risk pregnancy to improve the detection and monitoring of hypertension (BUMP) trials.[24] There was no direct patient involvement in the research reported here, as the focus was staff views and experiences. However, patient priorities were central to developing the trials and intervention development work reported elsewhere.[25]

**Table 1** Study sample

| Hospital | Focus group | Numbers | Staff represented |
|---|---|---|---|
| Guy's and St Thomas' NHS Foundation Trust | Focus group 1 | 10 | Obstetricians, midwives, pharmacist |
| | Focus group 2 | 4 | Day assessment unit midwives |
| | PI interview | 1 | Consultant obstetrician |
| West Middlesex University Hospital, Chelsea and Westminster Hospital, NHS Foundation Trust | Focus group 1 | 3 | Midwives |
| | Focus group 2 | 4 | Midwives |
| | Focus group 3 | 5 | Midwives and obstetrician |
| | Focus group 4 | 3 | Midwives |
| | PI interview | 1 | Consultant obstetrician |
| Whipps Cross Hospital, Barts Health NHS Trust | Focus group 1 | 11 | Obstetricians, midwives |
| | Focus group 2 | 4 | Antenatal midwives |
| | Focus group 3 | 4 | Day assessment unit midwives |
| | PI interview | 1 | Consultant obstetrician |
| The Royal London Hospital, Barts Health NHS Trust | Focus group 1 | 6 | Community midwives (FMU) |
| | Focus group 2 | 4 | Community midwives (in hospital) |
| | Interview | 1 | Consultant obstetrician |
| | Focus group 3 | 7 | Antenatal midwives |
| | Focus group 4 | 2 | Medical trainees |
| | PI interview | 1 | Consultant obstetrician |
| John Radcliffe Hospital, Oxford University Hospitals NHS Foundation Trust | Focus group 1 | 11 | Community midwives (in hospital) |
| | Focus group 2 | 3 | Obstetricians |
| | Interview | 1 | Obstetrician |
| | Interview | 1 | Consultant obstetrician |
| | Interview | 1 | Obstetrician |
| | Focus group 3 | 2 | Hospital midwives (MAU) |
| | Interview | 1 | Hospital midwife (MAU) |
| | PI interview | 1 | Consultant Obstetrician |
| Horton General Hospital, Oxford University Hospitals NHS Foundation Trust | Focus group 4 | 13 | Community midwives |
| Royal Berkshire NHS Foundation Trust | Focus group 1 | 2 | Hospital midwives |
| | Focus group 2 | 2 | Hospital midwives |
| | Interview | 1 | Hospital midwife |
| | Focus group 3 | 2 | Obstetricians |
| | Interview | 1 | Obstetrician |
| | Focus group 4 | 6 | Community midwives (in hospital) |
| | PI interview | 1 | Consultant obstetrician |
| The Royal Wolverhampton NHS Trust | Focus group 1 | 3 | Obstetricians |
| | Focus group 2 | 2 | Hospital midwives |
| | Focus group 3 | 10 | Hospital and community midwives |
| | Focus group 4 | 8 | Hospital and community midwives |
| | Interview | 1 | Hospital midwife |
| | PI interview | 1 | Consultant obstetrician |
| Birmingham Women's Hospital, Birmingham Women's and Children's NHS Foundation Trust | PI interview | 1 | Consultant obstetrician |
| | **Total** | **147** | |

FMU, free-standing midwifery unit; MAU, maternity assessment unit; NHS, National Health Service; PI, principal investigator.

## RESULTS

We heard views from a total of 147 healthcare professionals providing maternity care in the English National Health Service, of whom 37 were physicians (including principal investigators (PIs) and students) 109 midwives and one pharmacist (see table 1). Our findings suggest SMBP during pregnancy was not yet a widespread practice across the sites included.

**Table 2** Quotations

| Theme | Quotations |
|---|---|
| i) Interpreting BP changes in pregnancy | **BP Fluctuations**<br>'Even as midwives we get confused with what should be normal because we have a protocol that they have a booking [......measurement] so we've got a start and if it's 15 either way for the diastolic and we have to sort of think well yes that's creeping up a little bit if these girl's sent them to me I would look at it and say well that's not particularly high blood pressure because I'm dealing with PET all the time, you know, high blood pressure to me is 150/90 and above. So, isn't it, it's quite difficult and I think for a midwife let alone a woman to sit at home and think is that high am I going to be bothered' **Midwife**<br><br>'Another thing outside of pregnancy and not, most of what you'll doing with hypertension monitoring is trying to prevent long term complications on a short term and therefore, actually what you, slightly over 24 hours probably is what really matters whereas in pregnancy [....] I'm actually only interested in whether or not they've got [...] Well actually I'm really interested in the next 24 hours.' **Obstetrician**<br><br>**Wider symptoms**<br>'Yes but I think you raise a really important point about when you start to kind of focus on individual symptoms of a condition it's, there's a risk that you just focus on the blood pressure which means if you're feeling dizzy or your ankles are puffy or you're not really feeling yourself and got a blinding headache you don't focus on any those things because the blood pressure says it's okay.' **Community midwife**<br><br>'I think that they're over checking their blood pressure and occasionally I think it is helpful that they see a midwife and do things like check the protein in the urine and things like that which, you know, occasionally perhaps not be able to manage quite so well yourself' **Obstetrician**<br><br>'So you kind of think oh are we only doing half of a job if they're just monitoring their blood pressure' **Community Midwife** |
| ii) Reliability and accuracy | 'As a health professional I know blood pressure's affected by many things, whether you are taking it standing whether you are taking it lying whether it's in the morning whether you, or you've just come back from Tesco with plastic bags and stuff like, you're out of your breath or you've just fought with your partner or something like that, so many things influences blood pressure. So they will need to be given the empowerment they will need to take their blood pressure so that we, it's as accurate as, so that we get as close to the normal if that can be achieved, something like that.' **Midwife**<br><br>'I wouldn't ignore it if it was very elevated but I would certainly repeat it, I wouldn't go with what they said… Because I'd be worried about the quality of their machine' **Obstetrician**<br><br>'We advise them not to because [um] they're not calibrated, you know, they're their own there's no way they're calibrated and also quite a lot of [um] blood pressure [um] equipment, we've read research that says it's not as accurate as manual or a professional taking the blood pressure' **Midwife**<br><br>'When they come along and they bring readings [um] and they, if there's any conflict and sometimes they don't want to be labelled as hypertensive because it leads to certain choices later on in pregnancy that may be removed from them i.e. where they deliver, how they deliver [um] I then, you know, I will talk to them and say you, whatever you're using at home may or may not be validated for pregnancy but what we use here is validated for pregnancy and therefore I would defer to those readings' **Obstetrician**<br><br>'I tend to be a little bit sceptical because you don't know what they're using, you don't know what machine they're using, you don't know whether they're trained to use it, you don't know how old that machine is whether it's again cuff size, you know, if they've borrowed it off somebody you don't know whether it's been PAT tested, you know, and if they are concerned about their reading being particularly high or particularly low I'd rather just get them in to see the relevant person at the time rather than rely on that, you know, I do worry about them using their own equipment because you just don't know how accurate it is' **Midwife** |
| iii) Impact on women | 'So for the women who have had severe pre-eclampsia in the past who are absolutely panic stricken about it being missed in the future [......] I don't know scientifically if it matters but for them it gives them huge reassurance that they've got, it means that rather than them having to see the clinician every week or twice a week, they're doing something, it's rather like feeding a baby every other day.' **Obstetrician**<br><br>'There's a lot of women that have mildly raised blood pressure that are fine, that we don't really do anything with. [Um] and it's a lot of monitoring for them' **Obstetrician**<br><br>'Some women might not want to monitor themselves because they wouldn't feel, it will be too much responsibility in case they miss something' **Midwife**<br><br>'I think it's a bit of mix isn't it, I think some women would really like it saves them time especially if they're busy they have other children.' **Midwife**<br><br>'I think a lot of women would also appreciate it because they often don't want to come into hospital and they would, I think if they were self-monitoring at home if they're taught in the right way and they know when somethings high or they know when to escalate or to ring and I think it will work really well and stop a lot of people coming in unnecessarily. I think it would be really good' **Midwife** |
| iv) Anticipated impact on the antenatal care | 'We are so busy, we are too busy in hospitals and the more we can do out of hospital safely the better' **Obstetrician**<br><br>'There are benefits all-round aren't there, if you're, if we're [um] happy that this woman is monitoring her blood pressure sensibly [um] it's got benefits for her in that it means she doesn't have to come so often, it's got benefits for us in that it reduces our, our workload, surely it's got, it has a, there's an economic benefit there for the NHS [um] as a whole and it, I suppose that's one of the things isn't it that public health it's about educating people isn't it about, so that, you know, I think there are lots of benefits it's [um] it's just getting it in place. And being confident about the thing that you've got in place' **Midwife**<br><br>'I think it would have the potential to increase the appointments rather than decrease appointments as a screening tool for hypertension because of the worried well who've seen systolic increase in ten as an example but I don't think it will decrease the routine ante-natal screening because that has many other roles including measurement of fundal height or foetal wellbeing.' **Medical trainee**<br><br>'I suppose potentially it will increase workload because people are going to recognise that they have a slightly elevated blood pressure sooner but I don't think that's necessarily a bad thing' **Obstetrician**<br><br>'So it's a massive information imparting exercise probably more importantly than the, the systematic screening for obstetric complications that old fashioned people think of as ante-natal care. You know I, you know, again when I trained we'd see clinics of 40 people, 50 people in a morning where literally all you do would be check their blood pressure, check their urine, put your hand on their tummy, check their heart, you know, see if the patients still breathing, next, you know. But it isn't like that now there's a lot more education, seeing much more as a health education exercise as much as a screening for abnormality of pregnancy exercise.' **Obstetrician** |

Continued

**Table 2** Continued

| Theme | Quotations |
|---|---|
| v) Caution, uncertainty and evidence | **Caution**<br>'I wouldn't take the, take everything as a 100% as gospel because, you know, you can't really it's like triaging over the phone they can tell you so much but until you've seen that woman and made an assessment you can't do a full assessment from that. But [um] but yes I mean it would certainly I, I like compare to a jigsaw puzzle, you know, putting all the pieces together that would definitely be one of the pieces I would definitely include once I knew that all the bits were safe that they were using and everything, yes.' **Midwife**<br><br>'If I look at it and I can see that yes at 2:00 'o' clock and half two she had raised blood pressure but [….] guidelines about what I do about that [um] all I can do is do my own blood pressure readings and get them reviewed by the doctor. Unless there was like a real, you know, like a guideline of what we actually do about the trends, there's not much I myself could actually do from it.' **Midwife**<br><br>**Uncertainty**<br>'I think that would be very useful information to have rather than as you say the snap shot that you get in clinic. Although on the opposite side if they were having consistently low readings at home and when they come and visit us in triage it's always 160/110 I would question whether the data was true even if I knew that the machine was valid. Acknowledging the role of white coat hypertension but I would still be very worried about relying purely on the home readings and discarding the readings I have in the hospital.' **Medical trainee**.<br><br>'You can also have high blood pressure at home and lower blood pressure readings in hospital and we have no idea, I don't think [um] what we should do when actually readings are different in different places and there is a tendency, there will be some people who will have a tendency to treat anybody whatever the excuse, much more common is a tendency to say oh it's probably more, we've had 150, 140, 120 I'm going to believe the normal whether its home reading or hospital readings. And so just having more data may just be making more noise rather than giving us, telling us what to do. But without the data we don't even know if that's true.' **Obstetrician**<br><br>'Supposed to be evidence based and when you actually look at them about 90% of the recommendations they make are expert opinion' **Obstetrician** |
| vi) Concerns over action/inaction | 'Yes exactly and they may not realise the seriousness of it, so that would be my concern if they're doing it at home. The chronic hypertension women have a lot of contact with, in hypertension clinic they understand blood pressure they've been dealing with it, they're on medication they understand something about blood pressure, they might not fully understand how serious it can be with regards to pre-eclampsia but they certainly understand that they need to act on it and if they don't, obviously it's explained to them numerous times. My concern is if you did that to a general population will that message be translated.' **Midwife**<br><br>'I do have a worry about women taking, you know, something like pre-eclampsia is a multi-system disorder, blood pressure is one component and we see every week, every month women who come in with normal blood pressure but everything else going wrong and it would make me nervous about the idea of women taking control of their care so much so that they felt reassured by one reading and ignored their signs and symptoms and I think that, that will be, that's a major, that's a clinical worry that I have for using.' **Obstetrician**<br><br>'I think as long as they [women] understand the limits of what's normal and what's not and they know the right people to contact if not, then I don't see there's anything wrong with it' **Midwife**<br><br>'I just know that for some women they, they've got so much going on that actually their health is, is quite easily overlooked and [um] is last on the agenda if that makes sense and if they could miss a blood pressure reading here and there [um] then they may well do that and that would be my only [um] and which I feel awful saying really because I am so like passionate about actually women being responsible for their own care and taking ownership of it and us giving it back to them [um]. [……] I just think some women will be like oh yes, everything's fine they're all okay and maybe overlook things. I know you've got the opposite extent where some people would potentially act on it like say there's a higher reading because they think it will be acted on and that sort of thing as well but I think you could almost get women who will either pretend they've done it or pretend that it's an okay reading when it's not [um] which is awful to say that we don't trust women but' **Midwife** |

However, some midwives and obstetricians reported occasional pregnant women who were already self-monitoring, either of their own volition or because they had been advised to because of previous experiences of pre-eclampsia, WCH, or pre-existing hypertension. The main themes to emerge around SMBP were (1) different BP changes in pregnancy, (2) reliability and accuracy of BP monitoring, (3) anticipated impact of SMBP on women, (4) anticipated impact of SMBP on the antenatal care system, (5) caution, uncertainty and evidence, (6) concerns over action/inaction and patient safety.

### Interpreting BP changes in pregnancy
#### BP fluctuation
Staff emphasised that BP is different in pregnancy than at other times in women's lives. It fluctuates throughout pregnancy and these fluctuations are normal. However, BP in pregnancy can change rapidly, precipitating a medical emergency for mother and baby. While antenatal care staff will be familiar with this phenomenon through their training and experience, women may not be. There was concern women could have difficulty interpreting results and deciding when to seek acute care.

In the general population, BP readings are used to guide care and medication changes over the long term, but in pregnancy the situation is more dynamic. Some were interested in the wider context home BP readings would give them, in particular, feeling women with WCH might get more typical readings at home. However, several indicated they would tend to privilege their own clinic readings as they reflected the situation happening immediately in front of them.

#### Wider symptoms
Staff were also concerned that BP is not the only factor in diagnosing hypertension or pre-eclampsia but rather one piece of a diagnostic 'jigsaw'. Healthcare professionals stressed the importance of ensuring women were alert to the wide range of symptoms (eg, headaches, dizziness, blurred vision, proteinuria) and would act on those even if their BP appeared to be in the normal range.

## Reliability and accuracy

The reliability and accuracy of machines were key concerns for staff, as well as whether women would monitor their BP correctly and respond appropriately. Many raised concerns about the quality of home BP monitors, whether women were using one of the relatively few validated for use in pregnancy, and whether they were properly calibrated. There were concerns about appropriate cuff size, particularly for women with a high body mass index. Other factors identified as potentially impacting the reliability and accuracy of home readings included women's practices and the context in which they took readings (did they sit correctly, choose a quiet time of day?).

## Views on the impact on women

Staff had a range of perceptions about how pregnant women might feel about being asked to self-monitor their BP. Some anticipated women being happy to participate and enjoying a sense of agency, control and reassurance over their own healthcare. Others raised concern that women may experience increased anxiety and become obsessed about their BP. Some felt pregnant women expect care to be led by providers and asking them to monitor their own BP would be perceived as a way of abdicating responsibility. Although staff also acknowledged the potential for empowerment, some expressed concern that asking women to self-monitor would contribute to an over-medicalisation of pregnancy in contrast with the evolution of antenatal care away from this model in recent years.

Many potentially positive impacts for women were identified both in terms of quality of experience and better outcomes for mother and baby. These included fewer medical appointments, shorter waiting times and avoiding hospitalisation. Staff acknowledged detecting BP increases earlier would have the potential to deliver better outcomes for both mothers and babies, but confirmed definitive evidence of these outcomes would be needed before they could commit to changing their practice.

## Anticipated impact on antenatal care

Staff were caught between feeling SMBP could reassure women, so cutting 'worried well' visits and/or raise anxieties, thus increasing them. So, while they recognised other potential health system benefits including fewer women coming into clinic, thus freeing up clinic time in hospital and in community care, they also raised various concerns around the impact of SMBP on antenatal care pathways and their workload. These included a rise in phone calls to maternity assessment units and an increased demand for care, particularly from the 'worried well'.

Staff were aware that, ahead of definitive trial results, these reflections were hypothetical. Their currently available best comparator was their experience with women's self-monitoring for gestational diabetes, which is now quite common. Some described SMBP as a challenge to their professional roles, particularly for midwives, and part of wider changes towards increasing self-care and personal responsibility in health. One consultant described the shifting responsibilities in antenatal care, with health education now as much a part of routine care as systematic screening for obstetric complications.

## Caution, uncertainty and evidence

While staff were generally open to looking at home BP readings, most were mindful the evidence was not yet available as to how to treat these readings in conjunction with clinic readings. Most felt they could only treat home readings as additional rather than as core information, not directing care decisions, but contributing extra detail to the 'jigsaw'.

### Caution

Many midwives said that if self-monitoring readings were discrepant with clinic readings, they would privilege the readings they had just taken themselves. Most added they would take into account the home readings, though exactly how would be considered on a case-by-case basis. Some would use them as part of improving understanding of the wider context, but use their clinic reading for the decision on whether to admit a woman to hospital. The home reading would not be used for treatment decisions but could influence how often an obstetrician felt it necessary to see a woman.

Some acknowledged the potential of SMBP to identify WCH. But in several midwife discussions, consensus emerged on the side of caution, such as referring women onwards or taking bloods if only one type of reading (clinic or home) was high. Others referred to local protocols and guidelines although this was often interpreted as needing to act on any higher reading, implying caution and a focus on patient safety, with fear of being considered negligent explicitly raised by one midwife.

Some said they verified home readings by testing the device used by women alongside their clinic monitor. Several midwives commented they would have to respect a woman's readings, as to do otherwise could affect mutual trust.

### Uncertainty

There was awareness that pre-eclampsia is a variable condition in which the speed of disease progression varies considerably. Some felt SMBP could significantly mitigate some of these uncertainties. One senior obstetrician specifically alluded to a potential *advantage* of SMBP being its ability to reduce uncertainty by enabling clinicians to make objective assessment of how accurate clinic readings are, clarifying who has WCH, assessing the effectiveness of treatment and standardising the various clinical environments that women might be seen in. Yet others clearly feared SMBP could increase uncertainty (discrepancy in readings), and/or lead to overcaution and overtreatment on the part of less experienced colleagues.

Different approaches to managing hypertension in pregnancy also emerged. While some make judgements

on a case-by-case basis, privileging the extent of change from baseline, the wider context, and individualised pathways, and given that different hospitals may have their own guidelines, others described a practice defined by the National Institute for Health and Care Excellence guidelines and the use of thresholds as the defining guide in when to intervene.[26]

### Evidence

Numerous staff raised the need for additional and improved evidence. One obstetrician observed that guidelines are supposed to be evidence based but recommendations are often made on expert opinion. A different senior obstetrician also suggested that people practise defensively, leading to many unnecessary blood tests being done because a greater number of healthcare professionals (midwives and junior doctors) are able to request the tests, and because when women have been told they may have pre-eclampsia, heightened anxiety leads to repeated checks for reassurance. Overall, there was a distinct lack of consensus, but with a clear wish from many for more evidence to reduce uncertainty.

### Concerns over action/inaction

While a shift to SMBP is in keeping with wider trends towards self-care and personal responsibility, working through this new balance of responsibility for acting on readings was a cause for concern. Staff raised concerns about women not acting on readings, through either not realising they required urgent action, not wanting to bother busy staff or not wanting to be hospitalised. There were concerns about women ignoring problems, or being falsely reassured by one reading and ignoring the wider context. Many stressed the importance of women having a clear understanding of what they were supposed to do if their readings were high, with a clear pathway for women presenting with high BP based on home readings.

Many staff made comparisons with home monitoring of blood glucose in gestational diabetes, either as a useful precedent for SMBP or evidence of the challenges of women deliberately or inadvertently coming in with inaccurate results.

Others expressed the view that it is hospital policy (as well as being the health professional's view) that they would rather ask women to come in for comprehensive assessment than have the woman check her BP at home and decide not to come in.

### DISCUSSION
### Main findings

Antenatal care staff were generally in favour of asking women to monitor their own BP at home, but flagged concerns about the potential impacts on women and staff workload, the reliability and accuracy of readings and the need for clarity about how the information would be used by clinicians. Many emphasised the importance of ensuring women understood what to do in the event of

readings that cause concern and provided examples of other situations where women had failed to act in acute situations because they did not understand the severity of the situation or rationalised away the need to seek medical attention. Some voiced concerns about women becoming either anxious or falsely reassured by self-monitoring. There was a feeling that SMBP might only be for a select few as it will require a shift in thinking about who takes responsibility for the woman's health.

Many of the issues raised were specific to pregnancy. However, concerns about the potential for monitoring leading to anxiety may also exist for the general population, though previous studies have not found evidence to support this.[27] These could be intensified when there may be additional anxiety related to the effect of BP changes on the baby. Concerns about device accuracy were well founded as relatively few monitors on the market are validated in pregnancy.[28] The need to consider more than just BP and the dynamic nature of BP in pregnancy as opposed to BP in the general population means that failure to act has the potential for devastating consequences for both woman and baby. Yet, resistance to medicalisation may be heightened in a previously healthy younger population of 'women' as opposed to 'patients'.[29] Continuity of care is an ambition not always achieved in maternity care. The lack of an established relationship of trust between women/patients and healthcare professionals that is typical of hypertension management in primary care, further complicates the situation.[30] Staff are therefore understandably cautious and uncertain, while also willing to see the potential for SMBP to reduce uncertainty and add usefully to available evidence.

### Strengths and limitations

A strength of this study is the breadth of multidisciplinary voices across the maternity care pathway captured, spanning eight differently located hospitals in England. This has given rich insights into the potential challenges of introducing SMBP into maternity care from the perspectives of different disciplines and locales. Our two-stage approach to analysis, led by social scientists with expert clinical input from coauthors in obstetrics, midwifery and primary care, has also enhanced the interpretive strength of the findings. Limitations include the challenges of holding focus group discussions with busy professionals engaged with frontline care. These are vital perspectives to capture but there were inevitably challenges in terms of finding time and space for some focus group discussions.

There are several literatures to draw on in interpreting these results.

### Responsibilisation

To date, measuring and monitoring women's BP in pregnancy has been a central component of the healthcare professional's role in antenatal care. To include women's home readings into these care pathways represents a significant shift in the balance of responsibility. Emerging literature that explores the shifts and implications of health

'responsibilisation' are of relevance. Shifts in the balance of responsibility have the potential to disrupt long established care pathways and can be experienced both positively as well as negatively as surveillance.[31] Exploring the impacts of these shifts is a live issue in the sociological literature from studies of 'park runs' to HIV management,[32 33] and the relationships between social workers and vulnerable youth.[34] Drawing on work by Garrett, Liebenberg et al explored the impact of shifts in neo-liberal systems, where individuals are expected to manage their own risks and demonstrate self-care. 'These shifts impact on the roles that health professionals play, from active management to encouraging/teaching of the skills of management'.[35] Newman et al's study of young people with HIV revealed narratives of responsibilisation that can give rise to contest between young people and their clinicians.[33] In exploring the impact of telecare on the management of long term conditions, Rogers et al concluded 'Indeed a paradox of the reliance and acceptance of telecare is the creation of new relationships and dependencies rather than the diminution of reliance envisaged by policy.'[36] Our findings on clinician caution and concerns over inaction, indicate that these shifts in responsibility could have profound impacts that we need to be alive to as we further explore the impact of this work going forward.[37]

### Impact on professional roles

A shift in responsibility for BP monitoring may also have an impact on professional roles and identity. Concern over these impacts ran through various themes we have reported, including impact on workload, caution and uncertainty and location of expertise. Task shifting is a key issue in many healthcare settings, but it does not emerge seamlessly as revealed in the literature.[38–41] How SMBP impacts on these professional roles, and how professionals behave when there is a shift in the locus of responsibility will need to be the focus for future work.

### Patient safety

While staff raised concerns about the impact of SMBP on patient safety, these concerns were expressed in the context of an emerging evidence base for SMBP.[13 17] As this develops, we draw on evidence that points to the considerable scope for improving patient and family contributions to the detection and management of acute illness.[42] Delayed recognition and treatment of conditions such as pneumonia and meningitis in childhood,[43] pre-eclampsia and reduced fetal movements during pregnancy and after childbirth[44 45] and heart disease and stroke in adulthood,[46 47] contribute significantly to the mortality and morbidity burden. These conditions typically present with a time-critical window for early recognition and response, and are associated with red flag signs and symptoms (such as breathlessness and pain) which can signify a serious underlying condition and act as potential markers to aid patient and family involvement in escalation of care. However, research has shown there are challenges to speaking up and raising safety alerts in maternity care and that organisation-focused efforts are required to improve staff responsiveness.[48]

## CONCLUSIONS

We need to be mindful that the impact of SMBP on maternity services could be profound although nuanced. While its introduction will not reduce the responsibility clinicians have for women's health, it promises to enhance the responsibilities and agency of women, and introduces a new set of relationships into maternity care. We cannot simply look to the literature and experiences of SMBP in other populations (eg, such as hypertension outside of pregnancy) because of the very nature of BP changes in pregnancy and because this is a healthy population, unless specifically indicated otherwise. This is, then, a new space for reconfiguration of roles, mutual expectations, and the relationships between, and responsibilities of, healthcare providers and mothers.

**Author affiliations**
[1]THIS Institute, Department of Public Health and Primary Care, University of Cambridge, Cambridge, UK
[2]Primary Care Clinical Sciences, Institute of Applied Health Research, University of Birmingham, Birmingham, UK
[3]Nuffield Department of Primary Care Health Sciences, University of Oxford, Oxford, UK
[4]Freelance Researcher, Montreal, Québec, Canada
[5]Women's Health Academic Centre, King's College, London, UK
[6]Department of Midwifery and Child Health, City University of London, London, UK
[7]Department of Women and Children's Health, Kings College, London, UK

**Acknowledgements** We would like to thank all NHS staff who gave their time to contribute to this study.

**Contributors** LH, JH, KLT, LC, SG, CM, JS, LR and RJM conceived the study, secured the funding and were involved in the planning and carrying out the study. LH and JH conducted the interviews and focus groups and KLT managed the study. LH, JH and LR analysed the data with input from the wider authorship. LH and JH drafted the paper. All authors commented on drafts and writing up the work.

**Funding** This article represents independent research commissioned by the National Institute for Health Research (NIHR) Programme for Applied Research (RP-PG-0614-20005). Richard McManus was and Lucy Chappell is supported by a Research Professorship from the National Institute for Health Research (NIHR-RP-R2-12-015 and RP-2014-05-019, respectively). Richard McManus in an NIHR Senior Investigator. Lisa Hinton, Katherine Tucker and Richard McManus have received funding from the National Institute for Health Research (NIHR) Collaboration for Leadership in Applied Health Research and Care Oxford at Oxford Health NHS Foundation Trust and NIHR Oxford Thames Valley Applied Research Collaboration. Lisa Hinton was supported by the National Institute for Health Research (NIHR) Oxford Biomedical Research Centre (BRC). Jane Sandall is a National Institute for Health Research (NIHR) Senior Investigator and is supported by the National Institute for Health Research (NIHR) Applied Research Collaboration South London (NIHR ARC South London) at King's College Hospital NHS Foundation Trust.The views expressed in this publication are those of the author(s) and not necessarily those of the NHS, the National Institute for Health Research or the Department of Health.

**Competing interests** RJM has received BP monitors for research from Omron.

**Patient consent for publication** Not required.

**Ethics approval** A favourable ethical review for this study was obtained from South Central—Oxford C Research Ethics Committee (16/SC/0386). Written informed consent was gained from individuals who agreed to take part.

**Provenance and peer review** Not commissioned; externally peer reviewed.

**Data availability statement** Data are available upon reasonable request. The interview transcripts are held by the University of Oxford and available at reasonable request.

**ORCID iDs**
Lisa Hinton http://orcid.org/0000-0002-6082-3151
James Hodgkinson http://orcid.org/0000-0002-7583-5278
Katherine L Tucker http://orcid.org/0000-0001-6544-8066
Richard J McManus http://orcid.org/0000-0003-3638-028X

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
