## [Reviewer comments · BMJ Open]

ARTICLE DETAILS

TITLE (PROVISIONAL)	Exploring the potential for introducing home monitoring of blood pressure during pregnancy into maternity care: current views and experiences of staff. A qualitative study.
AUTHORS	Hinton, Lisa; Hodgkinson, James; Tucker, Katherine Louise; Rozmovits, Linda; Chappell, Dr Lucy; Greenfield, Sheila; McCourt, Christine; Sandall, Jane; McManus, Richard J

VERSION 1 – REVIEW

REVIEWER	Karen Tran University of British Columbia, Canada
REVIEW RETURNED	31-Mar-2020

GENERAL COMMENTS	The manuscript provides very key information regarding perception on the utility and limitations of self monitoring of blood pressure during pregnancy. The authors surveyed a wide range of clinicians and nursing staff. The responses and themes generated are very applicable and universal concerns that many clinicians have regarding implementing SMBP, including variability of blood pressure in pregnancy, reliability and accuracy of home BP monitors, and impacts on women and how care is delivered. Some areas for consideration: 1. To include demographics of the subjectives interviewed, % midwives, physicians, years of experience, age, etc into results.2. Question posed in Page 6, line 49 is unclear, consider rewriting3. Consider expanding on how participants were recruited and selected into the focus groups and interviews. Were this participants part of previous studies on using SMBP? Was sampling purposive vs. convenience sampling?4. Can you comment if ethics and informed consent was obtained for this study. It is not clearly identified in the text.5. Please consider referencing Table 1 in the results section of manuscript. Also, incorporating demographics of the participants surveyed would be helpful.6. Please consider reference Table 2 in the results section of manuscript.7. Some of the responses elicited in the manuscript mention staff vs. consultant vs. midwives. It would be important to clarify differences between staff vs. consultant (i.e.: nurses vs family physicians vs obstetricians), as the clinical role of training may offer different perspectives. It is unclear if additional analysis were done to see if responses differed between midwives vs. family physicians vs. obstetricians.8. A description of how maternity care is provided in London, where interviews are occurring would be helpful to understand how
---

	midwives and obstetricians work/manage women with pre-eclampsia. In my setting, midwives would not be looking after women with preeclampsia as they would be considered high risk. Perspective on how maternity care is delivered would help understand the flow of maternity care and work load.
--	---

REVIEWER	Richard Gentry Wilkerson University of Maryland School of Medicine Baltimore, Maryland United States of America
REVIEW RETURNED	16-Apr-2020

GENERAL COMMENTS	The submitted manuscript is a qualitative study using semi-structured interviews of clinicians at 8 hospitals that will be taking part in a multi-center research study regarding the use of self-monitoring of blood pressure in pregnancy. The purpose of the study was to explore perspectives and concerns about this process. Thematic analysis of the interviews demonstrated some main themes that are detailed in the paper. The manuscript was written in accordance with the SPQR checklist and all standards appear to have been addressed. Table 1 lists the hospitals and participants Table 2
--

REVIEWER	Hanis Hidayu Binti Kasim Universiti Sains Islam Malaysia (USIM), Malaysia
REVIEW RETURNED	18-Apr-2020

GENERAL COMMENTS	Dear Author, The result demonstrates high clinical importance that may be helpful to evaluate the affected areas for SMBP implementation in pregnant women, based on the maternity healthcare provider's perspective. Having a variety level of maternity staff and large subjects for a qualitative study is their major strength and the method used also replicable and feasible. The results are well discussed as categorized by the emergent themes. The part written as 'There are several literatures to draw on in interpreting these results' is immensely helpful for the readers to get a clear understanding, by connecting the method used with the results. The conclusion has answered their objective in a good manner and the references are adequate. There is one comment after reviewed the article, -The abstract is professionally written. Problems and objectives described precisely. However, as stated in the submission guidelines, the primary or secondary outcome measured should be included in the abstract. For this study context, the list of questions explored during the interview and focus group discussion, as written in the sub-title-data collection, from lines 7 to 19, should be included in the abstract (method), as it will be able to give the reader a quick view on the method used. Thank you.
--

REVIEWER	Anita Beelen Department of Rehabilitation, Physical Therapy Science & Sports, UMC Utrecht Brain Center, University Medical Center Utrecht, the
-----------------	--

	Netherlands Center of Excellence for Rehabilitation Medicine, UMC Utrecht Brain Center, University Medical Center Utrecht, and De Hoogstraat Rehabilitation, Utrecht, the Netherlands
REVIEW RETURNED	17-Jun-2020

GENERAL COMMENTS	General comments: This manuscript describes an interesting study using a qualitative approach to explore the perspectives of maternity staff to introducing and implementing SMBP in maternity care. The study follows up on earlier work exploring women’s experiences of taking part in an SMBP. For the present study the authors have included a very large sample of health care professionals working in maternity care. In studying the context and challenges to implementing SMBP in maternity care it would have been valuable to also include women receiving maternity care for their perspectives on SMBP (which are of utmost importance for implementation) but I realize that this was not the scope of this study. The manuscript is well written and the methodology was sound although the reporting needs some improvement as well as the reporting of results. More detailed comments (and suggestions for improvement) Abstract: add the concept of data analysis of the interviews and focusgroup to the methods Methods: Although the authors stated that the SPQR checklist was used when writing their report, several items are not addressed (and the included checklist refers to the wrong pages). The following aspects need to be reported in more detail: - Qualitative approach and research paradigm - Qualifications of the social science/nonclinical researchers - Data collection methods: triangulation of sources (interviews and focus groups). Please add the topic guide for individual interviews and for focus groups as an appendix - Data analysis: describe the process by which themes were identified and how trustworthiness of the thematic analysis was ensured. Describe how credibility, transferability, confirmability, and dependability was established Results: The naming of the identified themes overlap with the topics that are addressed in the interviews and most are too broad and insufficiently reflecting the data. E.g. the theme “Reliability and accuracy” was a topic of the interviews (“Questions sought to explore issues around.....the accuracy of home readings,”. Suggestion to rename it to “concerns with accuracy of home monitoring equipment”. Describe the meaning of the theme rather than the topic: e.g. the theme “ Different blood pressure changes in pregnancy” does not give the reader immediately a sense of what this theme is about. I presume (but lack expertise in maternity care) the theme should be somewhat like “interpreting changes in blood pressure”. Theme “Impact on women” does not capture the core findings on this topic. DATA TABLE 1: why are the (n=8) interviews with lead consultant obstetrician in hospitals (upper part of the table) separated from the interviews with other healthcare professionals (second half of the table, column Focus Group)? In the analyses there was no distinction made between these 2 groups. It would be better to include the 8 interviews with other HCPs in the upper part of the
--

	table, adding up to 16 interviews (and 131 participants in focus groups). Reporting checklist for qualitative study: the page numbers are incorrect (e.g. Ethical issues pertaining to human subjects are reported on page 24 (rather than 16).
--	---

VERSION 1 – AUTHOR RESPONSE

In response to comments from all four reviewers we have revised our manuscript as follows:

- (i) R1 requested that we include some demographics about the participants in our study. We are unable to provide the age or years of experience as this data was not collected, but are able to confirm that our of 147 there were 37 physicians (including PIs and students) 109 midwives and 1 pharmacist. We have provided an updated table with more detail on the breakdown and re-ordered the table, as per the comment from R4 and added a sentence to the start of the results.
- (ii) R1 raised a question about sampling and we have clarified that our sampling was convenience sampling. The phrasing of the question in paragraph 3 of the introduction has been rephrased for clarity. Participants had not taken part in previous studies.
- (iii) R1 asks us to clarify if the study had ethical approval. This is listed in supplementary detail, but we have included a sentence in the methods confirming that ethics and informed consent was obtained.
- (iv) We have provided additional reference to Table 1 in the Methods and Results section. Table 2 is already referenced in Methods and we feel does not need referencing again in the Results section.
- (v) R1 has asked for a description of how maternity care in provided in London. We have provided some more detail and a reference in the Methods section, on the multidisciplinary teams that provide care to women with additional risk factors in pregnancy in the English NHS, as not all hospitals were in London. We have also clarified that responses from midwives and obstetricians were analysed together, reflecting the multidisciplinary teams providing care.
- (vi) R3 has requested that we include primary and secondary outcome measures in the abstract. As this is a qualitative study, reporting outcome measures is not appropriate. However, we have included details of the qualitative nature of the enquiry in the abstract, to emphasise the design.
- (vii) R4 has requested that we add the concept of data analysis of interviews and focus groups to the methods and has requested some further detail on our approach to analysis and the qualifications of the social science researchers. These have been added to the Methods section. We have added the topic guide to our submission.
- (viii) R4 has suggested some renaming of the themes presented in the Results section.
 - (a) “Reliability and accuracy”. We disagree with R4’s suggestion that we re-name this theme “concerns with accuracy of home monitoring equipment”. While this was a sub-theme, we are reporting here wider concerns that include concerns about whether women will be able and willing to monitor their blood pressure reliably and accurately.

(b) We have however followed R4's next suggestion and renamed the theme on blood pressure changes, "Interpreting changes in blood pressure in pregnancy" and added further detail to the theme describing the impact on women, "Views on the impact on women".

- (ix) Formatting amendments: The corresponding author's details have been updated to reflect her new institution. The tables have been embedded in the text of the article.

VERSION 2 – REVIEW

REVIEWER	Karen Tran University of British Columbia, Canada
REVIEW RETURNED	04-Aug-2020
GENERAL COMMENTS	Thank you to the authors for your corrections. This study will add to a large gap of knowledge on SBPM in pregnancy and viewpoint of clinicians.